# Physical Activity and Sedentary Behavior in High School Students: A Quasi Experimental Study via Smartphone during the COVID-19 Pandemic

**DOI:** 10.3390/children10030479

**Published:** 2023-03-01

**Authors:** Regina Márcia Ferreira Silva, Lauryane Fonseca Terra, Michele da Silva Valadão Fernandes, Priscilla Rayanne E. Silva Noll, Alexandre Aparecido de Almeida, Matias Noll

**Affiliations:** 1Department of Education, Federal Institute Goiano, Ceres 76300-000, Brazil; 2Department of Education, Federal Institute of Tocantins, Araguatins 77950-000, Brazil; 3Health Science Graduate Program, Faculty of Medicine, Federal University of Goiás, Goiânia 74001-970, Brazil

**Keywords:** adolescents, exercise, lifestyle, schoolchildren, technologies

## Abstract

The objective of this study was to evaluate whether exposure to information about physical activity and its barriers can increase the level of physical activity and reduce the time exposed to sedentary behaviors in high school students involved in integrated professional and technological education during the coronavirus disease 2019 pandemic. This quasi experimental study was conducted with integrated education high school students, divided into two groups: Intervention Group (IG; n = 59) and Control Group (CG; n = 54). Physical activity and sedentary behavior were identified and measured using the International Physical Activity Questionnaire pre-and post-intervention for both groups. IG students received educational material thrice a week for four weeks. The focus of the material was the importance of physical activity and need to reduce the time exposed to sedentary behavior. The results revealed that IG students showed an average daily reduction of 47.14 min in time exposed to sedentary behaviors, while the CG students showed an increase of 31.37 min. Despite this, the intervention was not effective in improving physical activity levels in the IG and the mean reduction in the time exposed to sedentary behavior was not significant (*p* = 0.556). The intervention was ineffective in increasing the practice of physical activity and reducing the time exposed to sedentary behavior.

## 1. Introduction

Physical activity is a concept defined as any movement that results in energy expenditure above resting levels [1]. Recently, it has been extended to people moving, operating, and acting in culturally specific spaces and contexts, influenced by diverse interests, emotions, and relationships [2]. Regular physical activity has been considered a important factor associated with the prevention of chronic non-communicable diseases [3,4,5]. Additionally, studies have recognized physical, psychological, and social benefits associated with regular participation in physical activities [6,7,8,9,10,11,12,13,14,15,16]. Despite this, the prevalence of physical inactivity is above 80% among adolescents worldwide, when those who engage in (moderate to vigorous) physical activity for fewer than 60 min a day are considered physically inactive [17]. In Brazil, this percentage exceeds 83% for this age group [18].

Physical inactivity is characterized by failure to meet current physical activity recommendations [19,20,21], while sedentary behavior is defined as any waking behavior characterized by an energy expenditure ≤1.5 metabolic equivalents (METs); for example, time spent sitting or lying down while awake [20]. The factors that make it difficult or prevent individuals from participating in physical activities are commonly called barriers [22]; therefore, it is possible for a person to be physically active (by complying with the recommendations) and simultaneously spend significant time engaging in sedentary behavior. Barriers can contribute to an increase in physical inactivity and sedentary behaviors. An example of sedentary behavior is the use of electronic devices and prolonged sitting, lying down, or reclining [20]. 

The barriers can be categorized into four dimensions: environmental; psychological, cognitive, and emotional; sociodemographic; and sociocultural [23,24,25,26,27,28]. In this sense, the coronavirus disease 2019, a disease caused by the SARS-CoV-2 virus that spread into a worldwide pandemic [29], can be considered a barrier to the practice of physical activity, as it directly affected two important public health concerns: physical inactivity and sedentary behavior [30]. The COVID-19 pandemic impacted society in an unprecedented way, characterized by the greatest interruption of the teaching–learning process in the history of world education [31,32]. Thus, educational institutions worldwide needed to adopt emergency remote teaching modalities [33,34].

To promote physical activity, actions are needed that encompass concepts related to physical inactivity, sedentary behavior, and barriers to physical activity. Our study used the ecological framework theory of health behavior, which states that behavioral change is a complex process that can be elicited through a multi-pronged approach targeting intrapersonal, interpersonal, organizational, and socio-community dimensions [35,36]. In this sense, considering that adolescents engage in significant cell phone use during their day [37,38], it can be expected that “positive” messages related to healthy lifestyle habits and physical activity will contribute to a change in sedentary behavior observed in these young people. Subsequently, a line of reasoning highlights the possibility of extrinsic motivation bringing the individual closer to the activity, suggesting an increase in autonomy and an internal incentive [39].

Accordingly, the World Health Organization suggests the use of smartphones to help reduce time spent in sedentary behaviors and increase the practice of physical activity [5]. Recent systematic reviews, including articles of moderate to high quality, identified that smartphone-based physical activity interventions, such as those delivered via application, were effective in increasing an individual’s amount of physical activity [40,41]. Another review noted the need for studies that developed physical activity interventions using mobile health for specific target groups [42]. Several intervention studies used smartphones to promote a healthier lifestyle [43,44,45,46]. In a recent study [44], five weeks of using a mobile application that sent notifications about nutrition and physical activity to Portuguese adolescents improved diet behavior in 28.6% of participants. In addition, a 42.9% increase in the level of physical activity was observed. In another study [43], an application was used to apply individual and collective challenges related to physical activity to Spanish adolescents for 10 weeks. The results showed that the application increased participants’ time spent on physical activity. Finally, an application used for four weeks with female adolescents in Singapore attenuated the decline in physical activity level among participants [45]. Evidence showed that smartphone-based interventions may be a promising strategy for increasing total physical activity time [47,48] and reducing time exposed to sedentary behaviors [49] in adolescents.

Considering the above, an intervention that proposes increasing the level of physical activity and reducing exposure to sedentary behavior during the coronavirus pandemic could be an interesting strategy. Thus, this study aimed to assess whether exposure to information about physical activity and its barriers received via messaging applications can increase physical activity and reduce sedentary behaviors in high school students enrolled in professional and technological integrated education.

## 2. Materials and Methods

### 2.1. Study Design and Research Location 

This was a quasi-experimental study with a field trial design lasting four weeks. This study was conducted in September 2021 during the school year at a professional and technological education institution located in central Brazil. 

At that time, the population of Brazil had already been living with the COVID-19 pandemic for 18 months. The Midwest region of Brazil had the highest COVID-19 mortality rate in the country (301 deaths per 100,000 inhabitants), while the national rate was 249.9 deaths per 100,000 inhabitants [30,50]. Vaccine coverage at that time was just over 20% of the adult population [30,50].

The region was engaged in social isolation, with mandatory use of face protection masks. This state had the seventh lowest rate of social isolation in the country; however, the institution had been practicing emergency remote teaching for over a year [30,50]. 

### 2.2. Sample

Participants were recruited among 207 students enrolled in integrated technical courses offered by a federal public institution of professional and technological education. Everyone received the invitation to participate by e-mail. The participants belonged to five groups encompassing the first and third years of technical courses in agriculture, information technology, and the environment. We determined the quantitative sample using a 5.0% margin of error and a 95.0% confidence level, resulting in a sample of 113 participants. We used block randomization [51], which is useful to resolve imbalances in the number of individuals. Each of the five groups corresponded to a randomization block; these blocks were randomized into two groups. Of a total of 113 high school students, 59 belonged to the Intervention Group (IG) and 54 to the Control Group (CG).

### 2.3. Inclusion and Exclusion Criteria

For the inclusion criteria in the study, students should be properly enrolled in the institution. They should have a smartphone device with functionality and use of a text message application. Finally, they should have the cognitive ability to interpret and answer the questionnaires in the pre-and post-intervention periods. Students who responded inappropriately to questionnaires intended to collect information on these criteria were excluded.

### 2.4. Intervention

The general focus of the interventions was to improve knowledge about the importance of physical activity and reduce the time exposed to sedentary behaviors. The content of the intervention was based on the strategy developed by the group ‘On Your Feet Britain (10 ways to sit less at work)’ [52] and an intervention carried out with university students [53]. We focused on activities described in the Physical Activity Guide for the Brazilian Population designed for children and young people from 6 to 17 years of age in the domains of free time, displacement, school, and household chores [54].

Currently, mobile technologies are essential to human life, as they bring convenience and practicality to the touch of the screen [55]. A special highlight is the smartphone, which offers various applications that support activities such as study, work, and leisure, among others [56]. One such application is WhatsApp, which has become a widely used communication tool for personal relationships and professional activities [57].

The intervention included sending eight illustrated and colored folders [58] over four weeks. The folders were sent thrice a week through the WhatsApp messaging application. The participants were asked to “reply” to confirm when they had received and read our messages. All folders are included in this study’s supplemental material. Folders were sent on Mondays and Wednesdays, and both folders were resent on Fridays (see Table 1).

### 2.5. Data Collection

Data collection was performed using two questionnaires and a structured interview with open questions. The International Physical Activity Questionnaire (IPAQ), short version [59,60,61], was used to collect information on the level of physical activity and time exposed to sedentary behavior, and a 10-item questionnaire developed by the authors was used to collect information relevant to the research. The IPAQ was administered three days before and three days after the intervention to both participant groups. 

The questionnaire developed by the authors included 10 questions (satisfaction level with the project, language, content and terms; duration and number of questions; if they were encouraged to have a less sedentary week and encouraged to have a week with more physical activities; and score for the project from 0 to 10), and its function was to evaluate the intervention with open and closed answers. Therefore, its use occurred three days after the intervention only in the IG. The two questionnaires were conducted online via Google Forms. The interviews were conducted online with six participants selected from the IG. This amount is recommended for homogeneous samples [62]. Later, the interviews were transcribed and analyzed, and the results were grouped into thematic axes, categories, and indicators. 

All enrolled subjects voluntarily participated in this study (with parents’ consent and approval), following ethical principles. Our study was approved by the Ethics Committee of the Instituto Federal Goiano (No. 28163120.4.0000.0036).

### 2.6. Data Analysis

The level of physical activity was identified through questions 1–3 of the IPAQ and were classified as low, moderate, and high. The physical activity level in the participants was classified as “low” (not meeting the criteria for the “moderate” or “high” categories), “moderate” (at least 20 min of vigorous physical activity three or more days/week; at least 30 min of moderate physical activity or walking five or more days/week; or any combination of walking, moderate, or vigorous physical activity reaching at least 600 metabolic equivalents of task (MET) minutes/week at least five days/week), and “high” (vigorous physical activity reaching at least 1500 MET minutes/week at least three days/week; or any combination of walking, moderate, or vigorous physical activity reaching at least 3000 MET minutes/week at least seven days/week) [34,59].

The time exposed to sedentary behavior (minutes/day) was identified through question 4 of the IPAQ. It was determined from the weighted average of the time sitting on a weekday and a weekend day according to the following equation: [(weekday sitting time ∗ 5 + weekend day sitting time ∗ 2)/7]. 

We used descriptive and inferential statistics. Average daily exposure to sedentary behaviors in IG and CG before and after the intervention was calculated by delta; absolute delta was calculated by subtracting the average time exposed to sedentary behavior from the post-moment by the pre-intervention moment. The comparison of the mean was carried out with the *t*-test (SPSS 26.0).

To calculate the sample size, we used G-Power [63], with α = 0.05 (significance level) and β = 0.85 (power of the test) and found that the minimum sample of each group should be 31 participants. Content analysis was used to interpret qualitative data [64] from the interviews. All steps of this analysis were performed by two reviewers with experience in qualitative approaches. 

## 3. Results

The final sample consisted of 80 (26 female; 54 male) participants. After the four-week intervention period, 26 students had dropped out of the IG and three more dropped during the analysis stage. In Figure 1, based on the Consolidated Standards of Reporting Trials (CONSORT), we described the records of the intervention. Four participants dropped out of the CG. This is because they did not respond to the post-intervention questionnaire.

The age of the CG was 15.9 ± 1.15 years, and that of the IG was 16.2 ± 0.94 years (*p* = 0.225), others characteristics are found in the (Table 2).

The most frequent level of physical activity in the pre-and post-intervention of both groups was high (Table 3). In both groups, there was an improvement in the frequency of low and moderate levels. The high level remained at the same frequency before and after intervention in the IG and increased slightly in the CG.

Regarding the time exposed to sedentary behavior, a reduction in the mean time of 47.14 min per day in the IG and an increase in the mean time of 31.37 min per day in the CG was observed. However, no significant differences were observed in pre-or post-intervention means in both groups (Table 4). 

Table 5 presents data on the perception of IG students in relation to participation in the intervention. We also verified whether the intervention contributed in any way to the increase in the practice of physical activity and the reduction in the time exposed to sedentary behaviors of the participants. Through the responses to the intervention evaluation questionnaire, we identified that the majority (n = 24; 80.0%) demonstrated satisfaction in having participated in the project, with the language used and the duration. Only 10% (n = 3) of the participants did not make positive comments.

The interview responses revealed a need for future similar interventions to provide more options, such as sending notifications and inclusion of challenges, photos, and videos: 

“*I thought the project initiative was very good and really necessary, especially in the time of a pandemic where many stopped exercising, and with the project, people were encouraged to resume practice and even to start it for those who didn’t do anything before the pandemic. Great project, wonderful idea!!*”(P1)

“*The experience of this project was incredible because sometimes my routine and obligations end up distracting me from moving more, despite it being something I like. The project helped me to do this and motivated me in moments of fatigue. I loved participating.*”(P2)

“*Although I didn’t send photos or videos in the WhatsApp group, the messages reminded me to do some physical activity on days when I was very still. I found the project very interesting. I liked how it was done. Congratulations to those involved!*”(P3)

“*I liked receiving the messages and how the project was done in general; however, because I spend a lot of time on computers and in electronic games, almost without observing my notifications, actually, there were no changes in my routine.*”(P4)

“*I could see how long I sat while studying; I never stopped to think about it before.*” (P5)

The sedentary behavior axis was divided into five categories: concept, the total number of daily hours, number of daily hours (leisure), contributing factors, and positive factors related to the reduction in time exposed to sedentary behavior. In Table 6, the physical inactivity axis includes the knowledge category for the percentage of physically inactive Brazilian adolescents. 

In the thematic axis of sedentary behavior, the participants declared that they spent between seven and fifteen hours daily exposed to sedentary behavior (Table 6). Regarding the factors that contribute to sedentary behavior, they mentioned mainly three personal factors (lack of information, lack of interest, and laziness) and three external factors (pandemic, technology, and modern life). Regarding the positive factors caused by the decrease in sedentary behavior, they mentioned the increase in the disposition and prevention of diseases. In the thematic axis of physical inactivity, regarding the knowledge of the percentage of Brazilian adolescents who are physically inactive, the participants reported between 70% and 75%.

## 4. Discussion

The present study evaluated whether exposure to information about physical activity and its barriers could improve the level of physical activity and reduce the time exposed to sedentary behaviors in high school students enrolled in integrated professional and technological education. The high number of physically inactive adolescents [18] and the large number of health problems that exposure to sedentary behaviors can cause in general [65] demonstrate the need to evaluate efficient ways to reduce the time exposed to sedentary behavior and increase the level of physical activity. After the intervention, a mean decrease of 47.14 (min/day) in exposure to sedentary behavior was observed in the IG. In contrast, in the CG, we verified an increase of 31.37 min per day in the average time in the sedentary behavior activities. While the changes in time exposed to sedentary behavior were not statistically significant between pre-and post-intervention, the findings still have clinical relevance since “every step counts”, no matter how small the increase in physical activity is [5]. Regarding the practice of physical activity, no significant differences were identified between groups, demonstrating that they were homogenous groups in relation to basal condition (before intervention).

Two important factors could be contributed to this absence of significant differences. One was that, however motivational the messages sent to mobile applications were in increasing healthy behavior [66], adherence to social distancing and isolation strategies may have mitigated their effects. Another reason was the strong tendency to sedentary behavior during adolescence. Young adolescents who practice low levels of physical activity in early life tend to maintain low levels of physical activity practices in late adolescence and adult life [67]. As the COVID-19 pandemic was a totally unexpected situation, young people who did not practice adequate levels of physical activity before the establishment of measures to contain the transmission of the virus would not start the practice of physical activities during this period even if stimulated remotely, in this case, by application messages.

The messaging application was chosen first because it is the platform most reportedly used by adolescents between 15 and 17 years old. Approximately 70% of these young people use these technologies in their daily lives [37]. Another factor was the increase in the number of application downloads aimed at physical activity in the home during the pandemic [68,69]. A recent study observed the characteristics a smartphone application should contain to reduce sedentary behavior in adolescents. Among the characteristics mentioned, social relationships, messages, and updates were listed [70] as important. The mobile application used in the present study demonstrated good adherence by the sample analyzed.

The language used was simple and visually appealing; approximately 80% (n = 24) of the students reported being satisfied or very satisfied with the language, terms, and content used in the messages sent to them. These features are important as they arouse interest and facilitates use among the adolescent public. Although the format for disseminating the messages of the present study was well accepted or evaluated by the volunteers, more “relaxed” messages such as pictures or videos may receive more attention and greater dedication to the proposed changes in habits. Additionally, as the application was very popular among adolescents, there may have been a sharing of motivational messages between the experimental and control groups, confounding the interpretation of results. 

The duration of the intervention in this research is in line with a similar intervention [71]; both lasted four weeks. More than half of the students who participated in this research (n = 17; 56.6%) reported being satisfied or very satisfied with the duration of this intervention. An interventional study with university students for two, four, and six weeks showed that evidence, advice, reminders, and challenges sent by text messages have the potential to increase non-sedentary patterns [66,72]. Thus, similar interventions of longer duration may show significant effects.

Currently, technologies are essential to human life, as they bring convenience and practicality to the touch of the screen [55]. A special highlight among the technologies is the smartphone, which offers various applications that allow activities such as study, work, and leisure, among others [56]. A widely used application is WhatsApp, which has become a widely used communication tool for personal relationships and professional activities [57].

In research concerning high school students [66,73], similar to this study, an average of two hours a day in sedentary leisure-time behaviors was identified. In accordance with other studies [74,75], the use of technologies was reported as a factor that contributed to exposure to sedentary behavior. Motivating physical activity practices with smartphone or cell phone applications are conflicting. Although these applications may be effective to improve physical activity [76], a meta-analysis review found a non-significant effect [77]. In this sense, we may assume that an uncontrolled confounder in the study may have influenced our results. For example, cell phone use to access messages may stimulate adolescents to continue using their phones, thus increasing the time involved in sedentary behaviors.

To the best of our knowledge, this is the first study that investigated the use of mobile applications to influence increased sedentary behaviors in adolescents during an exceptional condition such as the COVID-19 pandemic. Furthermore, although our study sheds light on some important issues related to adolescent behavior (sedentary behavior and mobile use), it has some limitations. First, an important issue of the research design is that the sampling procedure did not ensure that the participants had same levels of PA experience before the lockdown period. However, in a real school context, it is not possible to evaluate classes of different intervention groups from the same school, to avoid communication bias. Second, we did not follow the post-intervention over time and the use of a self-reported questionnaire. Third, we cannot identify whether sedentary behavior continued to decrease or increase after the intervention. Fourth, during the pandemic, excessive use of the internet and social networking has been shown to contribute to an increase in depressive symptoms [34], which may have influenced the dropout of participants who possibly became “stressed” by so many online activities. We suggest that this be evaluated and considered in future studies. Finally, the lack of access to the data plan of some participants may have influenced the continuity of the intervention, which may have interfered with our results. Despite these limitations, given that knowledge surrounding adolescents’ physical activity during the lockdown and our study’s aim to shed light on the “unknown” period of the COVID-19 pandemic, we believe that the current study still made important advances in the literature targeting adolescents. Adolescents constitute an age group with high rates of sedentary behavior, and many interventions encounter problems in changing behaviors that hinder the increase of physical activity.

## 5. Conclusions

The intervention was not effective in increasing the practice of physical activity and reducing the time exposed to sedentary behavior in adolescent people. Some factors, such as frequency of sending messages, intervention time, content, and message formats, may have impacted the results and should be further investigated in future research. Future interventions should be improved with options beyond sending folders. We suggest exploring different aspects of feasibility. 

## Figures and Tables

**Figure 1 children-10-00479-f001:**
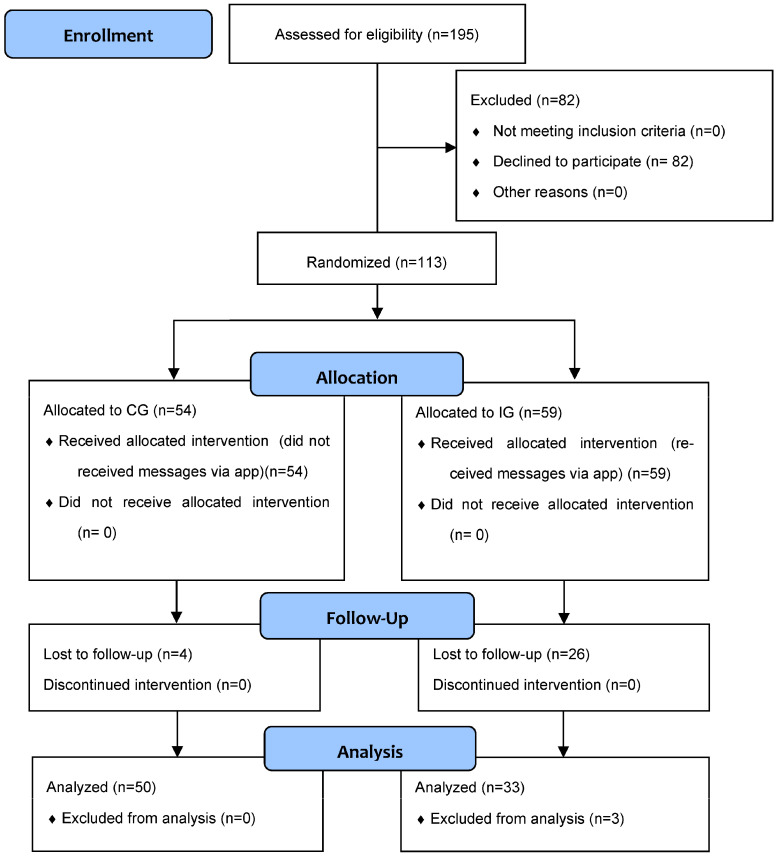
CONSORT (Consolidated Standards of Reporting Trials) 2010 Flow Diagram.

**Table 1 children-10-00479-t001:** The theme of the folders sent to the participants per week.

Week	Theme of the Folders
1st	Information and concepts related to physical activity and sedentary behavior. Suggestions for physical activities in the “free time” domain and two steps to reduce sedentary behavior.
2nd	Concepts and benefits of physical activity. Suggestions for physical activities in the “displacement” domain and two steps to reduce sedentary behavior.
3rd	Benefits and barriers to physical activity. Suggestions for physical activities in the “school” domain and two steps to reduce sedentary behavior.
4th	Benefits and dimensions of barriers to physical activity. Suggestions for physical activities in the “household chores” domain and two steps to reduce sedentary behavior.

**Table 2 children-10-00479-t002:** Frequencies by group, gender, and age of students.

		CG	IG
		n (%)	n (%)
Gender	Female (n = 26)	19 (38.0%)	7 (23.3%)
	Male (n = 54)	31 (62.0%)	23 (77.0%)

**Table 3 children-10-00479-t003:** Frequencies of physical activity levels in IG and CG (pre- and post-intervention).

Physical Activity Level		Pre-Intervention	Post-Intervention
		n (%)	n (%)
CG (n = 50)	Low	17 (34.0%)	13 (26.0%)
	Moderate	15 (30.0%)	18 (36.0%)
	High	18 (36.0%)	19 (38.0%)
IG (n = 30)	Low	9 (30.0%)	6 (20.0%)
	Moderate	5 (16.7%)	8 (26.7%)
	High	16 (53.3%)	16 (53.3%)

**Table 4 children-10-00479-t004:** Average daily exposure to sedentary behaviors in IG and CG before and after the intervention.

Group	Pre (min/day) (Mean ± SD)	Post (min/day) (Mean ± SD)	Absolute ∆ (min/day)	Relative ∆ (%)	* *p*
CG	485.46 ± 258.99	516.85 ± 272.05	31.37	6.46%	0.231
IG	541.71 ± 158.85	494.57 ± 142.27	−47.14	−8.70%	0.556

∆—delta; IG—WhatsApp message intervention group; CG—control group. The absolute ∆ was calculated by subtracting the average time exposed to sedentary behavior from the post-moment by the pre-intervention moment. * *t* test.

**Table 5 children-10-00479-t005:** Students’ perception of participation in the intervention.

Variables	Total 30 (100.0%)
Overall satisfaction level with the project	
Very satisfied	9 (30.0)
Satisfied	15 (50.0)
Neutral	4 (13.3)
Dissatisfied	0 (0.0)
Very dissatisfied	2 (6.7)
Satisfaction with language, content and terms	
Very satisfied	16 (53.3)
Satisfied	8 (26.7)
Neutral	4 (13.3)
Dissatisfied	0 (0.0)
Very dissatisfied	2 (6.7)
Satisfaction with the duration and number of questions	
Very satisfied	4 (13.3)
Satisfied	13 (43.3)
Neutral	11 (37.7)
Dissatisfied	0 (0.0)
Very dissatisfied	2 (6.7)
Encouraged to have a less sedentary week	
Yes	15 (50.0)
Partly	11 (36.7)
No	4 (13.3)
Encouraged to have a week with more physical activity	
Yes	16 (53.3)
Partly	10 (33.4)
No	4 (13.3)
Score for the project from 0 to 10	
0 to 3	0 (0.0)
4 to 6	0 (0.0)
7 to 8	13 (43.3)
9 to 10	17 (56.7)

**Table 6 children-10-00479-t006:** Thematic analysis of the participants: perceptions.

Thematic Axis	Category	Indicators
Sedentary behavior	Concept	- Not practicing physical activity - Sedentary person, remains seated or lying down for a long time
	Total number of daily hours in sedentary behavior	- 8 h to 10 h
	Number of daily hours in sedentary behavior (leisure)	- 1 h to 2 h
	Factors that contribute to sedentary behavior	- Lack of information - Lack of interest - Laziness - Pandemic - Technologies - Modern life
	Positive factors in reducing sedentary behavior	- More disposition - Disease prevention
Physical inactivity	Physically inactive Brazilian adolescents	- From 60% to 75%

## Data Availability

Additional data and SPSS software code can be obtained from the authors.

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
