# Peer review of "Physical Activity and Sedentary Behavior in High School Students: A Quasi Experimental Study via Smartphone during the COVID-19 Pandemic"

_children, 2023, doi:10.3390/children10030479_

Round 1

Reviewer 1 Report

This is a very interesting study which tries to shed light on the “unknown” period of the Covid-19 pandemic. Targeting adolescents is appropriate because it is an age group with high rates of sedentary behavior and many interventions encounter problems in changing these behaviors that hinder the increase of PA.

The study has some methodological issues; however, considering that our knowledge about PA of adolescents during the lockdown period is restricted, I strongly believe that the study should be published to inform the community. An important issue of the research design is that the sampling procedure could have ensured that the participants have different levels of PA experience before the lockdown period so that the researchers were able to determine the effect of the intervention on those with and without previous participation in PA.

Perhaps, giving more details about the failure of the attempt to ensure the above methodological assumption would have been necessary.

However, I understand the limitations of sampling during this period and do not consider this to be a substantial problem for the publication.

Author Response

January 28, 2023.

Manuscript Number: Children-2143871

Title: Practice of physical activity and time exposed to sedentary behavior in high school students: an educational intervention via mobile device during the COVID-19 pandemic

Dear Editor,

We would like to thank you, the Editor, and the Reviewers for the thoughtful and in-depth comments in our manuscript. Your suggestions and remarks have helped us to reflect on our paper and improve it. We appreciate your commitment and effort. We have carefully considered every comment, promptly accepted all the suggestions, and made the alterations as recommended using the red color in the manuscript.

Please find below a point-by-point response to the Editors’ and Reviewers’ comments with answers in red font.

Regarding Editor Comments:

We believe that the informed consent does not apply to our study. First, we would like to inform to the Editor that the manuscript does not have any image, initials or other personal information of the students participating in the educational intervention. Thus, the excerpts that bring the transcripts of the interviews with the participants are mentioned using only the letter "P" and is not possible to identify any participant. Second, the information is presented in the manuscript in a general and anonymous way, which contributes to the non-identification of the students participating in the intervention.  Moreover, this study was approved by the Research Ethics Committees of the Goiano Federal Institute and the Federal Institute of Goiás (no. 28163120.4.0000.0036 and no. 28163120.4.3001.8082).

Regarding the comment “Please reduce the repetition rate of your article while revising the manuscript. We would like to ask you to rephrase all the red marked places. We checked your manuscript and found that there are some sentences similar to former publications.” The 8% similarity with www.mdpi.com is due to the publication of another article by the same authors (https://doi.org/10.3390/su141911896) which is directly related to this one, since the two are the result of the first author's master's thesis. The relationships found are especially with regard to the names, attributions and institutions of the authors, references used, ethical approval and methodology of the instruments used. So, even most of all are not relevant, but we did a in-depth review, specially in our method section and rewrote the sentences, although several words are technical names.

REVIEWER 1

This is a very interesting study which tries to shed light on the “unknown” period of the Covid-19 pandemic. Targeting adolescents is appropriate because it is an age group with high rates of sedentary behavior and many interventions encounter problems in changing these behaviors that hinder the increase of PA.

The study has some methodological issues; however, considering that our knowledge about PA of adolescents during the lockdown period is restricted, I strongly believe that the study should be published to inform the community. An important issue of the research design is that the sampling procedure could have ensured that the participants have different levels of PA experience before the lockdown period so that the researchers were able to determine the effect of the intervention on those with and without previous participation in PA.

Perhaps, giving more details about the failure of the attempt to ensure the above methodological assumption would have been necessary.

However, I understand the limitations of sampling during this period and do not consider this to be a substantial problem for the publication.

Authors: Thank you so much for your positive feedback. We are glad to receive your comments and agree to the aforementioned limitations. Based on your comment, we include a sentence in the limitation section:

Lines 391-395: “…an important issue of the research design is that the sampling procedure did not ensure that the participants have same levels of PA experience before the lockdown period. However, in real school context, it is not possible to evaluate in the same from the same school classes different intervention group, in order to avoid communication bias.”

Based on your important comment, we also improved the discussion section with the following sentences:

Lines 403-409: “Despite these limitations, given that knowledge surrounding adolescents’ physical activity during the lockdown and our study’s aim to shed light on the “unknown” period of the COVID-19 pandemic, we believe that the current study still made important advances in the literature targeting adolescents. Adolescents constitute an age group with high rates of sedentary behavior, and many interventions encounter problems in changing behaviors that hinder the increase of physical activity.”

Reviewer 2 Report

First of all, we would like to thank the authors for reviewing this manuscript and congratulate them on their work.

The aim of the present investigation 

 was to evaluate whether exposure to information about physical activity and its barriers can increase the level of physical activity and reduce the time exposed to sedentary behaviors in high school students integrated with professional and technological education during the coronavirus disease 2019 pandemic, an interesting and current aspect in the field of sport and health sciences research.

Although the authors present a good description of the method, it is recommended that the Consort guide be used and that they indicate this in their paper.

The study design is not indicated in either the title or the abstract. The Consort guide indicates that it should be indicated in one or both places. It should be identified in the title or abstract.

The title is too long, although it clearly states the objective of the study, it is recommended that it be shortened. It could be shortened by stating: Physical activiy and sedentary behaviour....

Words are repeated in the title in the keywords. It is recommended to choose other words as keywords.

There is no argument as to why physical activity and avoiding sedentary behavior is important. It is recommended to include a paragraph describing the effect of physical activity and sedentary behavior on health.

The study design is not indicated in the Study design section. 

There is a lack of information on group randomization and blinding.

No indication of ethics committee or protocol registration in databases such as Clinical Trial.

The study sample size is not indicated.

In the Flow diagram 33 subjects analyzed in the intervention group are indicated, however, in the table the female and male add up to 30, as in the rest.

Indicating the age as over 16 and under 16 does not provide much information; also indicate the mean age and the range and standard deviation.

Author Response

January 28, 2023.

Manuscript Number: Children-2143871

Title: Practice of physical activity and time exposed to sedentary behavior in high school students: an educational intervention via mobile device during the COVID-19 pandemic

Dear Editor,

We would like to thank you, the Editor, and the Reviewers for the thoughtful and in-depth comments in our manuscript. Your suggestions and remarks have helped us to reflect on our paper and improve it. We appreciate your commitment and effort. We have carefully considered every comment, promptly accepted all the suggestions, and made the alterations as recommended using the red color in the manuscript.

Please find below a point-by-point response to the Editors’ and Reviewers’ comments with answers in red font.

Regarding Editor Comments:

We believe that the informed consent does not apply to our study. First, we would like to inform to the Editor that the manuscript does not have any image, initials or other personal information of the students participating in the educational intervention. Thus, the excerpts that bring the transcripts of the interviews with the participants are mentioned using only the letter "P" and is not possible to identify any participant. Second, the information is presented in the manuscript in a general and anonymous way, which contributes to the non-identification of the students participating in the intervention.  Moreover, this study was approved by the Research Ethics Committees of the Goiano Federal Institute and the Federal Institute of Goiás (no. 28163120.4.0000.0036 and no. 28163120.4.3001.8082).

Regarding the comment “Please reduce the repetition rate of your article while revising the manuscript. We would like to ask you to rephrase all the red marked places. We checked your manuscript and found that there are some sentences similar to former publications.” The 8% similarity with www.mdpi.com is due to the publication of another article by the same authors (https://doi.org/10.3390/su141911896) which is directly related to this one, since the two are the result of the first author's master's thesis. The relationships found are especially with regard to the names, attributions and institutions of the authors, references used, ethical approval and methodology of the instruments used. So, even most of all are not relevant, but we did a in-depth review, specially in our method section and rewrote the sentences, although several words are technical names.

REVIEWER 2

First of all, we would like to thank the authors for reviewing this manuscript and congratulate them on their work.

The aim of the present investigation was to evaluate whether exposure to information about physical activity and its barriers can increase the level of physical activity and reduce the time exposed to sedentary behaviors in high school students integrated with professional and technological education during the coronavirus disease 2019 pandemic, an interesting and current aspect in the field of sport and health sciences research.

Authors: Thank you so much for your positive feedback.

Although the authors present a good description of the method, it is recommended that the Consort guide be used and that they indicate this in their paper.

Authors: Thanks for your careful review. We followed The Consort Guide and we included a sentence on page 5 of the manuscript and the Flow Diagram (Figure 1).

The study design is not indicated in either the title or the abstract. The Consort guide indicates that it should be indicated in one or both places. It should be identified in the title or abstract.

Authors: Based on your suggestion, we have included the study design both on title and abstract:

TITLE: “Physical activity and sedentary behavior in high school stu-dents: an educational intervention via smartphone during the COVID-19 pandemic”

Line 15 (ABSTRACT) “This educational intervention was conducted with integrated high school students, ...”

The title is too long, although it clearly states the objective of the study, it is recommended that it be shortened. It could be shortened by stating: Physical activiy and sedentary behaviour.…

Authors: We accepted you suggestion, thanks. New title:

“Physical activity and sedentary behavior in high school stu-dents: an educational intervention via smartphone during the COVID-19 pandemic”

Words are repeated in the title in the keywords. It is recommended to choose other words as keywords.

Authors: Thanks for your suggestion. To meet your suggestion all keywords have been changed to

Keywords: “adolescents; exercise; lifestyle; schoolchildren; technologies”

There is no argument as to why physical activity and avoiding sedentary behavior is important. It is recommended to include a paragraph describing the effect of physical activity and sedentary behavior on health.

Authors: Thanks for your careful review. To addressed your comment, we included the following paragraph in lines 33 to 40:

“Regular physical activity has been considered a important factor associated with the prevention of chronic non-communicable diseases [3–5]. Additionally, studies have recognized physical, psychological, and social benefits associated with regular participation in physical activities [6,7,16,8–15]. Despite this, the prevalence of physical inactivity is above 80% among adolescents worldwide, when those who engage in (moderate to vigorous) physical activity for fewer than 60 minutes a day are considered physically inactive [17]. In Brazil, this percentage exceeds 83% for this age group [18].”

The study design is not indicated in the Study design section.

Authors: Thanks for your suggestion. On line 101, the study design is presented:

“This was an educational intervention study with a field trial design lasting four weeks.

There is a lack of information on group randomization and blinding.

Authors: Thanks for the observation. It was really not possible to blind, in an attempt to minimize this situation, we present this situation as a limitation of our study. This limitation is present in lines 391 to 395:

“First, an important issue of the research design is that the sampling procedure did not ensure that the participants had same levels of PA experience before the lockdown period. However, in a real school context, it is not possible to evaluate in the same from the same school classes different intervention group, to avoid communication bias.”

No indication of ethics committee or protocol registration in databases such as Clinical Trial.

Authors: Thanks for your careful review. Ethical approval is present in lines 167 to 169:

“All enrolled subjects voluntarily participated in this study (after parents’ consent and approval), following the ethical principles. Our study was approved by the Ethics Committee of the Instituto Federal Goiano (nº. 28163120.4.0000.0036).”

The study sample size is not indicated.

Authors: Thanks for the observation. We present the sample size in lines 120-121 of section 2. Material and Methods, as follow:

“Of a total of 113 high school students, 59 were part of the Intervention Group (IG) and 54 of the Control Group (CG).”

In the Flow diagram 33 subjects analyzed in the intervention group are indicated, however, in the table the female and male add up to 30, as in the rest.

Authors: Thanks for your careful review. This is due to the fact that three were excluded from the analysis step. This information can be verified in the last rectangle of the Flow diagram on page 5.

Indicating the age as over 16 and under 16 does not provide much information; also indicate the mean age and the range and standard deviation.

Authors: Based on your comment, we removed the information from the table and include a sentence, as follow:

Lines 241-242: “The mean age of the CG was 15.9 ± 1.15 years, and that of the IG was 16.2 ± 0.94 years (p=0.225)”.

Reviewer 3 Report

The paper explores an intervention implemented during the COVID-19 pandemic to increase physical activity and reduce sedentary behaviour. A major limitation of this study is the lack of acknowledgment or consideration to the context in which the study was conducted: during a global pandemic. The study does not consider nor explain movement/lifestyle restrictions that were in place during data collection that will have influence participants’ ability to be physically active. It is not clear whether the intervention factored in such restrictions when providing guidance for how to be more physically active and less sedentary. For example, in the 3rd week of the intervention the theme was physical activities for the school domain, but were children even allowed to go to school at this time? If this study is to have impact, such as providing insight into how interventions should be delivered during a global pandemic, participants should have been asked questions related to the acceptability and feasibility of carrying out the intervention under government/pandemic-related restrictions. The discussion and conclusion of the study do not consider the context of the pandemic.

Introduction

1.     Paragraph 1 and 2: It would be beneficial within this section to provide details of what current physical activity guidelines are, and therefore, what is meant when someone is classified as physically inactivity (i.e., failure to meet these guidelines)

2.     Lines 42-43: please revise the definition of sedentary behaviour so that is matches with the current recommended definition by Tremblay et al.

3.     Lines 41-42: The authors should add details regarding how sedentary behaviour and physical activity are viewed as distinct behaviours and that someone can be physically active but also highly sedentary and that this can have its own health implications

4.     Lines 45-46: this sentence regarding sedentary behaviour appears out of place and should follow on from the initial definition of sedentary behaviour at the start of the paragraph

5.     Lines 45: Please change to ‘time exposed to’

6.     Lines 47-48: It is unclear what the four dimensions are, since five factors are then listed

7.     Lines 48-55: This section does not logically follow the previous information. Suddenly information shifts to COVID-19. A clearer link is needed. In addition more background information should be provided regarding the COVID pandemic and why it became a barrier to physical activity, such as lockdown restrictions/Government regulations that were imposed.

Materials and Methods

8.     Section 2.1.: Since the study is framed around the COVID-19 pandemic, information is needed regarding the status of the pandemic when the study was conducted, such as restrictions that were in place that may influence physical activity and sedentary behaviour

9.     Line 126: It is not clear how participants ‘demonstrated’ they had read the message

11..  Lines 138-143: More information is needed regarding the questionnaire developed by the authors including the questions that were asked and how participants were permitted to respond (e.g., open-answer, closed-answer)

12..  Lines 142-143: more details are needed regarding the transcription and analysis of the interviews such as software used and how thematic/ content analyses were conducted. Also, who conducted the interviews and how they were conducted (e.g., online or in person)

Results

12.  Line 177: a significant number of participants dropped out of the IG. Can the authors provide reasons for this?

13.  Table 3: Please report data to a meaningful number of decimal places

Discussion

14.  Lines 304-305: How do the authors know this reduction in time has clinical relevance? Data/reference are needed to support such a statement

15.  Lines 351-356: This paragraph seems out of place for the discussion. Instead, justifying the use of WhatsApp and smart phone technologies would be better placed in the methods

16.  Limitations around the use of self-report physical activity and sedentary behaviour need to be acknowledged

Author Response

January 28, 2023.

Manuscript Number: Children-2143871

Title: Practice of physical activity and time exposed to sedentary behavior in high school students: an educational intervention via mobile device during the COVID-19 pandemic

Dear Editor,

We would like to thank you, the Editor, and the Reviewers for the thoughtful and in-depth comments in our manuscript. Your suggestions and remarks have helped us to reflect on our paper and improve it. We appreciate your commitment and effort. We have carefully considered every comment, promptly accepted all the suggestions, and made the alterations as recommended using the red color in the manuscript.

Please find below a point-by-point response to the Editors’ and Reviewers’ comments with answers in red font.

Regarding Editor Comments:

We believe that the informed consent does not apply to our study. First, we would like to inform to the Editor that the manuscript does not have any image, initials or other personal information of the students participating in the educational intervention. Thus, the excerpts that bring the transcripts of the interviews with the participants are mentioned using only the letter "P" and is not possible to identify any participant. Second, the information is presented in the manuscript in a general and anonymous way, which contributes to the non-identification of the students participating in the intervention.  Moreover, this study was approved by the Research Ethics Committees of the Goiano Federal Institute and the Federal Institute of Goiás (no. 28163120.4.0000.0036 and no. 28163120.4.3001.8082).

Regarding the comment “Please reduce the repetition rate of your article while revising the manuscript. We would like to ask you to rephrase all the red marked places. We checked your manuscript and found that there are some sentences similar to former publications.” The 8% similarity with www.mdpi.com is due to the publication of another article by the same authors (https://doi.org/10.3390/su141911896) which is directly related to this one, since the two are the result of the first author's master's thesis. The relationships found are especially with regard to the names, attributions and institutions of the authors, references used, ethical approval and methodology of the instruments used. So, even most of all are not relevant, but we did a in-depth review, specially in our method section and rewrote the sentences, although several words are technical names.

REVIEWER 3

The paper explores an intervention implemented during the COVID-19 pandemic to increase physical activity and reduce sedentary behaviour. A major limitation of this study is the lack of acknowledgment or consideration to the context in which the study was conducted: during a global pandemic. The study does not consider nor explain movement/lifestyle restrictions that were in place during data collection that will have influence participants’ ability to be physically active. It is not clear whether the intervention factored in such restrictions when providing guidance for how to be more physically active and less sedentary. For example, in the 3rd week of the intervention the theme was physical activities for the school domain, but were children even allowed to go to school at this time? If this study is to have impact, such as providing insight into how interventions should be delivered during a global pandemic, participants should have been asked questions related to the acceptability and feasibility of carrying out the intervention under government/pandemic-related restrictions. The discussion and conclusion of the study do not consider the context of the pandemic.

Authors: Thanks for the observation. The observation is extremely relevant, and such weaknesses are justified by the lack of knowledge of the entire context of covid in which we found ourselves at the time of the research. To try to clarify the context including the following excerpt in the Methods section:

Lines 104-111:

            “At that time, the population of Brazil had already been living with the COVID-19 pandemic for 18 months. The midwest region of Brazil had the highest COVID-19 mor-tality rate in the country (301 deaths per 100,000 inhabitants), while the national rate was 249.9 deaths per 100,000 inhabitants [30,50]. Vaccine coverage at that time was just over 20% of the adult population [30,50].

            The region was engaged in social isolation, with mandatory use of face protection masks. This state had the seventh lowest rate of social isolation in the country; however, the institution had been practicing Emergency Remote Teaching for over a year [30,50].”

Introduction

  1. Paragraph 1 and 2: It would be beneficial within this section to provide details of what current physical activity guidelines are, and therefore, what is meant when someone is classified as physically inactivity (i.e., failure to meet these guidelines)

Authors: Thanks for the observation. To addressed your suggestion, we have included the following excerpt at the end of the first paragraph:

Lines 37-40:  “...when those who engage in (moderate to vigorous) physical activity for fewer than 60 minutes a day are considered physically inactive [17].”

  1. Lines 42-43: please revise the definition of sedentary behaviour so that is matches with the current recommended definition by Tremblay et al.

Authors: Thanks for the observation. In Lines 42-43 We include the current definition in the second paragraph “sedentary behaviour is defined as any waking behaviour characterized by an energy expenditure ≤1.5 metabolic equivalents (METs)...”

  1. Lines 41-42: The authors should add details regarding how sedentary behaviour and physical activity are viewed as distinct behaviours and that someone can be physically active but also highly sedentary and that this can have its own health implications

Authors: Thanks for your in-depth review. To meet your suggestion, we include the following sentence:

Lines 46-48: “...therefore, it is possible for a person to be physically active (by complying with the recommendations) and simultaneously spend significant time engaging in sedentary behavior.”

  1. Lines 45-46: this sentence regarding sedentary behaviour appears out of place and should follow on from the initial definition of sedentary behaviour at the start of the paragraph

Authors: Thanks for your careful review. As requested, we updated the definition of sedentary behavior in the sentences before the one mentioned. Therefore, this sentence is just citing some examples of sedentary behavior, so we decided to only mention examples so that the text does not contain repeated definitions of sedentary behavior.

  1. Lines 45: Please change to ‘time exposed to’

Authors: Thanks for the observation. In line 72 We changed the term ‘time exposed to’ to “time spent in sedentary behavior”.

  1. Lines 47-48: It is unclear what the four dimensions are, since five factors are then listed

Authors: Thanks for the observation. We agree that the writing was confusing, so we've made some readjustments to make it easier to understand. the confusion is due to the fact that the second dimension has a name composed of three words "Psychological, cognitive and emotional". So we rewrite it as follows in lines 51-52:

“The barriers can be categorized into four dimensions: 1environmental; 2psychological, cognitive and emotional; 3sociodemographic; and 4sociocultural [23–28].”

  1. Lines 48-55: This section does not logically follow the previous information. Suddenly information shifts to COVID-19. A clearer link is needed. In addition more background information should be provided regarding the COVID pandemic and why it became a barrier to physical activity, such as lockdown restrictions/Government regulations that were imposed.

Authors: Thanks for the observation.

Lines 52-56 “In this sense, the coronavirus disease 2019, a disease caused by the SARS-CoV-2 virus that spread into a worldwide pandemic [29], can be considered a barrier to the practice of physical activity, as it has directly affected two important public health concerns: physical inactivity and sedentary behavior [30].”

Materials and Methods

  1. Section 2.1.: Since the study is framed around the COVID-19 pandemic, information is needed regarding the status of the pandemic when the study was conducted, such as restrictions that were in place that may influence physical activity and sedentary behaviour

Authors: Thanks for the observation. The observation is extremely relevant, and such weaknesses are justified by the lack of knowledge of the entire context of covid in which we found ourselves at the time of the research. To try to clarify the context a little we include the following excerpt in lines 104-111:

“At that time, the population of Brazil had already been living with the COVID-19 pandemic for 18 months. The midwest region of Brazil had the highest COVID-19 mor-tality rate in the country (301 deaths per 100,000 inhabitants), while the national rate was 249.9 deaths per 100,000 inhabitants [30,50]. Vaccine coverage at that time was just over 20% of the adult population [30,50].

The region was engaged in social isolation, with mandatory use of face protection masks. This state had the seventh lowest rate of social isolation in the country; however, the institution had been practicing Emergency Remote Teaching for over a year [30,50]. ”

  1. Line 126: It is not clear how participants ‘demonstrated’ they had read the message

Authors: Thanks for your careful review. To improve understanding we replaced the word “demonstrate” with “reply” in line 145.

11..  Lines 138-143: More information is needed regarding the questionnaire developed by the authors including the questions that were asked and how participants were permitted to respond (e.g., open-answer, closed-answer)

Authors: Thanks for your careful review. We rewrote this sentence in order to better describe the questions in the questionnaire developed by the authors in lines 157-161:

“The questionnaire developed by the authors included 10 questions (satisfaction level with the project, language, content and terms; duration and number of questions; if they were encouraged to have a less sedentary week and encouraged to have a week with more physical activities; and score for the project from 0 to 10), and its function was to evaluate the intervention with open and closed answers.”

I take this opportunity to mention that this information can also be found in table 4.

12..  Lines 142-143: more details are needed regarding the transcription and analysis of the interviews such as software used and how thematic/ content analyses were conducted. Also, who conducted the interviews and how they were conducted (e.g., online or in person)

Authors: Thanks for the observation. In section 2.5 we included the information that the interviews were conducted online. And in section 2.6, in its last paragraph, there is the analysis used for the qualitative data, as shown below in lines 193-195:

“Content analysis was used to interpret qualitative data [62] interviews. All steps of this analysis were performed by two reviewers with experience in qualitative approaches.”

Results

  1. Line 177: a significant number of participants dropped out of the IG. Can the authors provide reasons for this?

Authors: Thanks for the observation. Unfortunately, we do not know the reasons or factors that may have led to a large loss of participants in the intervention group. Therefore we include the following limitation

Lines: 397-401:

“Fourth, during the pandemic, excessive use of the internet and social networking has been shown to contribute to an increase in depressive symptoms [34], which may have influ-enced the dropout of participants who possibly became "stressed" by so many online ac-tivities. We suggest that this be evaluated and taken into account in future studies.”

  1. Table 3: Please report data to a meaningful number of decimal places

Authors: Thanks for the observation. As requested we standardized all data with two decimal places.

Discussion

  1. Lines 304-305: How do the authors know this reduction in time has clinical relevance? Data/reference are needed to support such a statement

Authors: Thanks for your careful review. We rely on the WHO assertion that “every step counts”, so we include the following excerpt in lines 328-331:

 “...While the changes in time exposed to sedentary behavior were not statistically significant between pre-and post-intervention, the findings still have clinical relevance since "every step counts", no matter how small the increase in physical activity is [5].”

  1. Lines 351-356: This paragraph seems out of place for the discussion. Instead, justifying the use of WhatsApp and smart phone technologies would be better placed in the methods

Authors: Thanks for the observation. As suggested we transferred the mentioned paragraph to the methods section.

Lines 137-142: “Currently, mobile technologies are essential to human life, as they bring convenience and practicality to the touch of the screen [55]. A special highlight is the smartphone, which offers various applications that support activities like study, work, and leisure, among others [56]. One such application is WhatsApp, which has become a widely used communication tool for personal relationships and professional activities [57].”  

  1. Limitations around the use of self-report physical activity and sedentary behaviour need to be acknowledged

Authors: Thanks for your careful review. This limitation was included in the last paragraph of the discussion session.

Line 396: “and the use of a self-reported questionnaire”.

Round 2

Reviewer 2 Report

Thank you for the review of the article but some recommendations have not been properly taken into account.

The information on sample size and power is not adequate.

The research design is not adequate. The designs could be descriptive, randomized controlled trial, cross-sectional, etc.

Author Response

February 3, 2023.

Manuscript Number: Children-2143871

Title: Practice of physical activity and time exposed to sedentary behavior in high school students: an educational intervention via mobile device during the COVID-19 pandemic

Dear Editor,

We would like to thank you, the Editor, and the Reviewers for the thoughtful and in-depth comments in our manuscript. Your suggestions and remarks have helped us to reflect on our paper and improve it. We appreciate your commitment and effort. We have carefully considered every comment, promptly accepted all the suggestions, and made the alterations as recommended using the red color in the manuscript.

Please find below a point-by-point response to the Editors’ and Reviewers’ comments with answers in red font.

REVIEWER 2

Thank you for the review of the article but some recommendations have not been properly taken into account.

The information on sample size and power is not adequate.

The research design is not adequate. The designs could be descriptive, randomized controlled trial, cross-sectional, etc.

Authors: Thanks for your careful review. To address your observation we have included more information about the study design in the title, abstratc (line 15) and method (line 101), as follow:

Line 15: “This is a quasi experimental study ...”

Line 101: “This was a quasi experimental study ...”

We have also included more information about the sample size in item “2.2 Sample” (lines 113, 117-119), as follow:

Line 113: “Participants were recruited among 207 students…”

Line 117-119: “We determined the quantitative sample using a 5.0% margin of error and a 95.0% confidence level, resulting in a sample of 113 participants. ”

Reviewer 3 Report

Thank you for taking the time to revise your manuscript based on the reviewer feedback. All comments made have been addressed. 

Author Response

February 3, 2023.

Manuscript Number: Children-2143871
Title: Practice of physical activity and time exposed to sedentary behavior in high school students: an
educational intervention via mobile device during the COVID-19 pandemic

REVIEWER 3
Thank you for taking the time to revise your manuscript based on the reviewer feedback. All comments
made have been addressed.
Authors: Thanks for your careful review. We thank you for the care and all the observations that contributed
a lot to a significant improvement of our manuscript.